

# Visual analytics of the aftershock point cloud data in complex fault systems

Chisheng Wang[1,3], Junzhuo Ke[1], Jincheng Jiang[2], Min Lu[1], Wenqun Xiu[4], Peng Liu[5], Qingquan Li[1]

[1]Guangdong Key Laboratory of Urban Informatics, School of Architecture & Urban Planning, Shenzhen University, Shenzhen, 518000, China
[2]Shenzhen Institutes of Advanced Technology, Chinese Academy of Sciences, Shenzhen, 518000, China
[3]Key Laboratory of Urban Land Resources Monitoring and Simulation, Ministry of Land and Resources, 518000, China
[4]Shenzhen Urban Public Safety and Technology Institute, Shenzhen, 518000, China
[5]Academy for Advanced Interdisciplinary Studies and Department of Earth and Space Sciences, Southern University of Science and Technology, Shenzhen 518000, China

*Correspondence to*: Jincheng Jiang (jiangjincheng0305@126.com)

**Abstract.** Aftershock point cloud data provide direct evidence for the characteristics of underground faults. However, there has been a dearth of studies using the state-of-art visual analytics methods to explore the data. In this paper, we present a novel interactive visual analysis approach for visualizing the aftershock point cloud. Our method employs a variety of interactive operations, rapid visual computing functions, flexible display modes and various filtering approaches to present and explore the desired information for the fault geometry and aftershock dynamics. The case study conducted for the 2016 central Italy earthquake sequence shows that the proposed approach can facilitate the discovery of the geometry of the four main fault segments and three secondary fault segments. It can also clearly reveal the spatio-temporal evolution of the aftershocks, helping to find the fluid-driven mechanism of this sequence. An open-source prototype system based on the approach is also developed and is freely available.

## 1 Introduction

Fault complexity of earthquakes is universal in all types of faults (strike-slip, normal, and reverse). Usually, the faults are segmented along the strike by geometrical discontinuities where they are divided into a number of subparallel segments with the lengths of approximately 10-25 km [*Scholz*, 1997]. In some cases, the depth segmentation of a fault is also observed [*J R Elliott et al.*, 2011] where the cascading earthquake occurs successively along the up-dip or along the down-dip. The geometrical discontinuities in the complex fault structure can sometimes impede and stop the rupture propagation and limit the earthquake magnitude, and in some cases act as a high-permeability conduit to allow fluid migration and control the spatio-temporal sequence of the subsequent earthquakes [*Walters et al.*, 2018]. The elastic rebound theory based on the earthquake cycle concept is commonly applied for long-term earthquake prediction. However, the complex fault structure complicates the seismic hazard assessment because theoretical models based on simplistic assumptions (e.g. single dynamic rupture) were found to be inapplicable to natural faults [*Barbot et al.*, 2012]. The time intervals for cascading fault rupture



due to fault segmentation range from seconds to years on a case-by-case basis, while the control factors of the faults are still not clearly understood. Therefore, it is highly important to understand the complex fault geometry and such understanding may provide useful insight into fault mechanics and help to constrain the existing theoretical models.

Due to the past lack of ability to conduct high-resolution observations, it is always difficult to define the geometry of the active fault system. Recent developments of both geodetic and seismic observation techniques make it possible to reveal the fault geometry at the surface and depth. Advanced SAR geodesy provides near-field deformation and can be used to directly map the fault surface and constrain the deep fault through deformation inversion. The construction of an increasing number of seismic stations allows the identification of the locations of a large number of aftershocks following the mains shocks.

The point cloud of the located aftershocks contains the information regarding both the three-dimensional coordinates and precise time of occurrence that can directly reveal the fault geometry and temporal evolution of the earthquake sequence. Current interpretation methods for aftershock cloud are mostly case-specific rather than general, and to date, no standard processing procedure for visualization of these data has been available. While many studies plot the cross-section profile of the aftershock point cloud to reveal the dip of the fault plane [*J Elliott et al.*, 2013; *Hengesh and Whitney*, 2016; *Irmak et al.*,

2012; *Li et al.*, 2010], these simple visualization methods usually have the following limitations. 1) Lack of interactive operations. The visualization methods provide a static display without an interactive interface to assists the users to perform effective analyses through various human-computer interactions. 2) Lack of visual analysis. The current methods simply present these data without computing the feature (e.g. plane fitting or local outlier factor value calculation) to enhance the structure. 3) Lack of noise filtering. While the aftershock locations normally contain many uncertainties, most studies do not

consider either the identification or removal of these noisy points. As was shown in many recent studies, aftershock data can reveal much more information if these data are presented well, and this is particularly true for a complex fault system [*Walters et al.*, 2018]. Therefore, it is necessary to develop an interactive visual system to effectively mine information from the aftershock data.

Visual analytics is an emerging field that integrates new computational and theory-based tools with innovative interactive techniques and visual representations to enable human-information discourse. This approach is highly advantageous and has good potential in complex data analytics for intuitive information representation and dynamic data exploration [*Ward et al.*, 2015]. The importance of scientific visualization has been recognized for a long time [*McCormick*, 1988]. Specifically, in geophysics, the visualization of seismic refraction data has already been widely applied to explore the distribution of

petroleum and gas [*R Liu et al.*, 2018; *Raposo et al.*, 2016]. However, to date, studies using the state-of-art visual analytics method to explore other kinds of seismic data (e.g. aftershock point cloud) to gain information about earthquakes have been lacking.



In this study, we propose a novel interactive visualization method for exploring the 3D aftershock point cloud data. This method can help researchers to better understand the complex fault geometry in three ways: 1) A set of interactive operations is provided to stimulate creative analysts and exploit the researchers' background knowledge; 2) On-the-fly computation of the local outlier factor (LOF), 2D projection and plane fitting are supported in the visual analysis so that the hidden features of the fault structure can be discovered dynamically; 3) The noisy or irrelevant aftershock points can be filtered to reduce the distractions from invalid information.

## 2 Visual analytics of aftershock data

Figure 1 shows the flowchart of the proposed approach. Initially, we imported the aftershock point data from an earthquake catalog or literature into our system. At the same time, the LOF values for all of the points are computed automatically. Then, the points are presented in a 3D view window, allowing free rotation, panning and zooming to observe the point structure. The earthquake fault, either from an existing source or from inferred information, can be imported to the 3D view as well. After defining the parameters for the projection (i.e. strike and dip of the plane, and azimuth of the projection direction), the 2D points are rapidly computed and presented. Then, some interactions, i.e. coloring and filtering, can be performed on the 3D point clouds and 2D plane projection to enhance the structure. From the 2D plane projection, two interactions are further provided for exploring the fault geometry and aftershock dynamics, respectively. The first is the selection and plane fitting of a point cluster on a potential plane. After interactively selecting the clustered points, a plane will be automatically fitted by a robust estimator to reveal the potential fault structure. The second is the observation of aftershock migration. Animation of the aftershocks is provided so that a preliminary recognition of the migration direction can be obtained. Then, the user can define a propagation direction to further explore the aftershock evolution. After assigning the direction, the distance traveled by the aftershocks along the specified direction as function of time will be plotted, which is helpful for discovering the driving factor of aftershocks migration. We note that when the preliminary exploration of the fault geometry or earthquake dynamics are finished, the users can return to the previous interactions to refine their interpretation of the data.

### 2.1 Visual Design

As shown in Figure 2, the user interface includes four main components. The first component is the operation tools panel with all buttons, sliders, textboxes and labels. It includes most interactions in the visual analytics. The second component is the 3D view panel for presenting cloud points and fault planes. The third component is the 2D view panel for presenting the points in the 2D projection plane. The fourth component is a temporal panel that switches between 3D fault plane fitting plotting and propagation distance-time plotting, corresponding to the explorations of the fault geometry and aftershock dynamics, respectively.





## 2.2 Interactive operations

The easy-to-use interactions represent a significant difference between the proposed method and conventional aftershock interpretation methods. The traditional static displays prevent the analyst from directly and rapidly exploring the aftershock data. However, the easy-to-use interactive displays can lead to deeper insight into 3D aftershock cloud point data and
discover new fault structure or spatial-temporal evolution pattern. These interactive operations are described as follows:

**3D View Manipulation**: The point cloud data can be displayed in a 3D view control (Figure 2b). Some typical 3D interactions are available including zooming, panning, rotating, data tips, and data brushing (Figure 2a), allowing interactive exploration and editing of the plotted data to improve the visual display.

**Interactive Projection**: The cross-section plot is the main visualization approach for exploring the complex fault geometry (Figure 2c). The faults are normally represented by linear features when the aftershocks are projected onto a plane perpendicular to the fault. We offer an interface to input the strike and dip of the projection plane for rapid plotting of a desired plane projection profile.

**Dynamic Filtering**: Noisy or irrelevant data can significantly affect the interpretation of the aftershock point cloud. The filtering or denoising interaction is an important step for removing noisy or irrelevant aftershock points while retaining the fault geometry details. The filtering is based on the four features of the aftershocks, namely the depth, LOF, time, and magnitude (Figure 2a). LOF is an index for finding anomalous data points that will be introduced below [*Breunig et al.*,
2000]. Filtering based on LOF is used to denoise the data.

**Color Switching**: The feature for coloring the aftershock points can be quickly switched between the depth, LOF, time, and magnitude, allowing the users to effectively understand the data from different dimensions (Figure 2a).

**Fault Plane Fitting**: To test the probability of an identified fault, the users first brush the points that form a linear feature in the cross-section plot (Figure 2c). A fault plane will be automatically fitted to these points based on a robust estimator (Figure 2d). This interaction combines expert knowledge and visual computing, allowing the user to quantitatively locate the potential fault structure.

**Fault Data Fusing**: Users can import the faults to the 3D view in order to assist the visualization and analysis of the aftershock point cloud (Figure 2b). These faults can be hypothetic seismogenic faults or background fault system.




**Animation**: The aftershocks can be plotted gradually to form an animation in both the 3D view and the 2D cross-section plot (Figures 2b and 2c). Animated visualization has unique advantages and aiding the viewers to interpret the evolution of the aftershocks, making comparisons between different points in time, chunking the aftershock points into smaller datasets, focusing attention, and anticipating and understanding the spatio-temporal changes in the aftershocks.

**Propagation distance-time plotting**: To further quantify the migration process of the aftershocks, users can define a propagation direction (Figure 2c) and plot the distance along this direction as a function of time (Figure 2d). The temporal trend may reveal the mechanism (e.g. fluid diffusion) underlying the aftershock migration.

### 2.3 LOF calculation

Because of the limitations due to the resolution of seismic data inversion, there is some uncertainty in the location of the aftershocks, so that some earthquake points show large deviations from the fault plane. Usually, the original aftershock data do not contain any information about this uncertainty, and we cannot eliminate these outliers. In this study, we proposed to use the LOF which is commonly used in the identification of the outliers in the LiDAR point cloud [*Wang et al.*, 2019] to detect the anomalous points that deviate distinctly from the fault plane. The LOF [*Breunig et al.*, 2000] is based on the

concept of local density, which is estimated by the reachability distance of the object to its k nearest neighbors. Points with a substantially lower density than their neighbors are considered outliers. By comparing the local reachability density of the object to those of its neighbors, its LOF value can be obtained. The LOF is calculated as follows:

Let kd(p) be the k-distance of a point p, defined as the distance from point p the k-th nearest neighbor. The reachability

distance between point p and o is then given by,

$$rd_k(p, o) = \max\{kd(o), d(p, o)\} \tag{1}$$

where d(p, o) is the actual distance between p and o.

The local reachability density of an object p is then defined by,

$$lrd_k(p) = \frac{|N_k(p)|}{\sum_{o \in N_k(p)} rd_k(p, o)} \tag{2}$$


where $N_k(p)$ is the k nearest neighborhood, defined as the set of points within the distance of the k-distance (p).

The LOF for point p is then obtained by,

$$LOF_k(p) = \frac{\sum_{o \in N_k(p)} lrd_k(o)}{|N_k(p)| lrd_k(p)} \tag{3}$$





Figure 3 shows an example of the calculated LOF values where an LOF value significantly larger than 1 indicates a sparse region.

**2.4 2D Projection**

Projecting the 3D aftershock point cloud to a 2D plane can significantly enhance the fault geometry features. As mentioned above, we offer an interactive way to rapidly compute the projection when a plane and projection direction are assigned. To make the interactive operation user-friendly for geologists, we use geological terms, i.e. strike and dip, to define the orientation of the projection plane, and azimuth angle to define the projection angle. We firstly calculate 3D geographic coordinates of the projected points on the projection plane for aftershocks according to the Line–plane intersection equation.

The 3D geographic coordinates are then transformed to 2D local coordinate system by two element rotations, a z-axis rotation by an angle of strike-90, and a x-axis rotation by an angle of -dip, as shown in Figure 4a.

The rotation matrices ($\mathbf{R}_z$ and $\mathbf{R}_x$) are given by,

$$\mathbf{R}_z(\text{strike} - 90) = \begin{bmatrix} \cos(\text{strike} - 90) & -\sin(\text{strike} - 90) & 0 \\ \sin(\text{strike} - 90) & \cos(\text{strike} - 90) & 0 \\ 0 & 0 & 1 \end{bmatrix} \quad (4)$$

$$\mathbf{R}_x(90 - \text{dip}) = \begin{bmatrix} 1 & 0 & 0 \\ 0 & \cos(-\text{dip}) & -\sin(-\text{dip}) \\ 0 & \sin(-\text{dip}) & \cos(-\text{dip}) \end{bmatrix} \quad (5)$$

The projected coordinates ($\mathbf{P}$) from the aftershock point ($\boldsymbol{u}$) is given by,

$$\mathbf{P} = \mathbf{R}_x \mathbf{R}_z \boldsymbol{u} \quad (6)$$

where the x- and y- coordinates of $\mathbf{P}$ correspond to the coordinates in the projection plane, and the z-coordinate represents the distance to the plane.

**2.5 Plane fitting**

When the analyst identifies a potential linear feature from the cross-section plot, the plane fitting can rapidly evaluate the possible feature and provides the precise geometric parameters. In this study, we used an algorithm based on the singular value decomposition to fit a plane to the given 3D aftershock points. The calculation of the parameters of the plane (ax+by+cz=d) based on the singular value decomposition (SVD) method is carried out as follows.

Give a set of points ($x_i, y_i, z_i, i = 1 \dots n$) and their center point ($\bar{x}, \bar{y}, \bar{z}$), we form a matrix A as given by,



$$A = \begin{bmatrix} x_1 - \bar{x} & y_1 - \bar{y} & z_1 - \bar{z} \\ x_2 - \bar{x} & y_2 - \bar{y} & z_2 - \bar{z} \\ x_3 - \bar{x} & y_3 - \bar{y} & z_3 - \bar{z} \\ . & . & . \\ . & . & . \\ x_n - \bar{x} & y_n - \bar{y} & z_n - \bar{z} \end{bmatrix} \tag{7}$$

We applied an SVD to matrix A

$$A = UDV^T \tag{8}$$

The parameter vector [a, b, c] is just the smallest singular vector corresponding to the least singular value, and the last parameter d can be obtained by,

$$d = a\bar{x} + b\bar{y} + c\bar{z} \tag{9}$$

To avoid disturbance from noisy points, we include an iterative step to identify outliers and exclude them from plane fitting.

This step is carried out using the following algorithm,

1) Fit a plane using the available points;

2) Calculate the distance (**D**) of the points to the plane;

3) Calculate the root mean square error (σ) of all of the distances (**D**);

4) Find points with distances 3 times larger than σ and define them as outliers;

5) Remove these points and then return to step 1;

6) Stop when there is no outlier found, and return the plane parameter.

Given the normal vector of the fitted plane (a, b, c), the strike and dip of the fault plane are given by (Figure 4b),

$$strike = atan2(sign(c) \times a, sign(c) \times b) \tag{10}$$

$$dip = acos(c/\sqrt{a^2 + b^2 + c^2}) \tag{11}$$

where $atan2(Y,X)$ is the four quadrant arctangent of the elements of X and Y, $-\pi <= atan2(Y,X) <= \pi$, and sign(.) is the

signum function that returns 1 for a positive element and -1 for a negative element.

**2.6 Prototype system**

The proposed visualization approach has been implemented in a software package (Aftershock Visualization [AFV1.0], https://github.com/caigenszu) and is freely accessible to the scientific community. This prototype system was developed in Matlab with Graphical User Interface (GUI), providing point-and-click control for all operations. An analyst with no

programming background does not need to learn a new language or type commands to run the application. Meanwhile, the source code of the package is also provided so that advanced users can easily read the data and results throughout the visualization processing, and can modify the package to include their custom-built modules. This package requires an input file consisting of the aftershock point cloud table in text format with each row represent a point, and the columns representing utm-x, utm-y, depth, magnitude, time (days from reference time), LOF, and event flag (whether or not the shock




is the main shock) in sequence. An alterative input file is the fault plane used to assist data interpretation. Each fault plane is represented by one row, and the columns represent strike, dip, center utm-x, center utm-y, length, upper-depth and bottom-depth.

## 3. Case study: the 2016 central Italy earthquake sequence

On 24 August 2016 (UTC 01:36, local time 03:36), a destructive earthquake (Mw 6.2) occurred in central Italy (the Amatrice earthquake). The epicenter was located at 42.70°N, 13.23°E, between the towns of Norcia and Amatrice. Two months later, two major earthquakes were triggered to the northwest of the Amatrice earthquake on 26 October (19:18 UTC) and 30 October (19:18 UTC), with individual moment magnitudes of 6.1 (Visso earthquake) and 6.6 (Norcia earthquake). The earthquake sequence occurred across several fault systems (Figure 5), including the Mt. Vettore–Mt. Bove Fault system

(VBFS) and the Mt. Gorzano Fault system (GFS). The VBFS is a SW-dipping fault system, composed of different splays and segments. The fault system scarps are exposed along the SW foothills of Mt. Vettore, Mt. Porche, and Mt. Bove, with a length of ~27 km. The GFS is a SW-dipping fault, with a ~26 km long fault scarp developing along the foothills of Mt. Gorzano. These two faults are segmented by existing tectonic structures inherited from pre-Quaternary compressional tectonics [*Pizzi and Galadini*, 2009]. A ~3 km thick layer in which small events and some large extensional aftershocks

occur is found below the seismogenic fault, limiting the seismicity to the first 8 km of the crust [*Chiaraluce et al.*, 2017].

Many recent studies reported in the literature have revealed a very complex fault geometry for this earthquake sequence. In addition to the main range-front normal faulting structures that were adopted in many early studies [*C Liu et al.*, 2017; *Tinti et al.*, 2016], recent studies find increasing evidence for some additional minor antithetic and synthetic faults [*Cheloni et al.*,

2017; *Chiaraluce et al.*, 2017; *Scognamiglio et al.*, 2018; *Walters et al.*, 2018]. The activation of this secondary fault structure has important implication for evaluating the seismic hazard in this section of central Apennines. The complex fault geometry also plays an important role in channeling fluid diffusion, and affects the spatial order of the cascading earthquakes. The aftershock distribution provides direct evidence for the fault geometry and underground dynamics. By interactively visualizing these data, we can obtain useful information regarding both the fault geometry and earthquake dynamics.

Additionally, a relocated aftershock dataset has been provided in the work of *Chiaraluce et al.* [2017], making this earthquake sequence an ideal case for applying our proposed interactive visualization method. Below, we demonstrate the benefits of our method in identifying fault geometry and aftershock migration.

### 3.1 Fault geometry identification

According to previous studies, several fault segments are involved in this earthquake sequence. Some of these are the main

fault segments located in the GFS-VBFS fault system while others are secondary fault segments including an NE dipping normal fault antithetic to the GFS-VBFS fault system, and a preexisting compressional structure that is likely related to a



segment of the Sibillini Thrust. We will show how our interactive visualization method can facilitate the recognition of these structures.

Following the flowchart shown in Figure 1, we first import the relocated aftershock data from the results obtained by
*Chiaraluce et al.* [2017] to the 3D view. Some of the basic interactions provided by this package, i.e. zooming, panning, rotating, data tips, and data brushing, can be used to explore the data. LOF values were then calculated for all points to identify the aftershock outliers. We can interactively select to show some of the points according to their location, depth, magnitude, LOF, and time, and project them to a desired plane. After we found a linear feature in the projection plane, we select the points comprising the linear feature and automatically fit a plane to the feature (Figure 6).

First, we identify the four main fault planes by projecting the cloud point to a plane with $70^0$ strike and $90^0$ dip. This plane is approximately perpendicular to the GFS-VBFS fault system, and is should reveal the cross-sections of the fault plane well. To better illuminate the linear features of the faults, we filter the aftershock points according to the fault location and the spanning time of the aftershocks on the fault. To denoise the point cloud, the aftershocks with the LOF value larger than 2
are filtered. We then selected the points around the linear feature for each fault plane, and automatically obtained the suggested geometry parameters of the fitted plane as shown in Table 1, finding that the obtained parameters are in overall agreement with the main seismogenic fault planes in previous studies.

Second, we explore the point cloud to find other secondary faults. Since there is no hint for the existence of these faults, we used the basic interaction tools in 3D view and the filtering function in our package to find the possible fault geometry. An
obvious cluster of aftershock points is found on an antithetic NE dipping plane in the Norcia area. We therefore also project the cloud point to a plane with $70^0$ strike and $90^0$ dip. After filtering the points by location and LOF value, we obtain a very strong linear fault signal as shown in Figure 6. Fault fitting gives a strike $-38^0$ of and dip of $59.8^0$ for the Norcia antithetic fault. However, other fault structures revealed in the literature are not very obvious in the aftershock cloud point. We then project the point cloud to a simplified seismogenic fault (strike=160, dip=50) along a horizontal vector with azimuth 15,
using the projection settings from *Walters et al.* [2018] . We filter the points by the magnitude, location, timing and LOF and then, two linear structure were observed as shown in Figure 6. The structure on the right is the NW dipping Pian Piccole fault that generates a significant surface displacement signal in the InSAR measurements. The structure on the left is an east-dipping fault that was only inferred in *Walters et al.* [2018]. We picked the points around these linear structures for plane fitting and obtain a strike of $-114^0$ and dip of $46.5^0$ for the Pian Piccole fault, and a strike of $7.8^0$ and dip of $85.3^0$ for the
inferred fault.

### 3.2 Migration of aftershocks

Besides the static analysis of aftershocks to find the fault geometry, the timing information of aftershocks have potential to reveal the earthquake dynamics such as fluid diffusion. Fluid diffusion is a main factor in triggering seismicity as observed



in many tectonic regimes [*Vidale and Shearer*, 2006]. It was also proposed to have been present in the nearby 2009 L'Aquila earthquake [*Malagnini et al.*, 2012]. The migration of aftershocks provides a direct evidence for such diffusive process. The migration of the aftershocks provides direct evidence for such diffusive process. *Walters et al.* [2018] found a significant fluid-driven aftershock migration between the Amatrice earthquake and Visso earthquake controlled by the fault intersection
in this earthquake sequence.

To observe the spatio-temporal process, we project the aftershocks to the seismogenic fault along a horizontal vector with azimuth 15 as described above. The points are colored by their rupture time. As shown in Figure 7a, we can observe a trend of the color change from green to red along the northward direction, implying the clear directionality of the aftershock
migration. We then used the animation operation to plot the aftershock points step by step as function of time in both the 2D projection plane and 3D view. The directional propagation of the aftershocks can be observed much more clearly.

Next, we can define a starting point and aa assumed direction in the projection plane. After drawing the start point and the propagation line, the distances of the aftershocks to the starting point along the direction can be calculated automatically.
They are then plotted as a function of time to further reveal the time evolution of the aftershocks. Following the work of *Walters et al.* [2018], we defined a starting point in the peak slip of the Amatrice earthquake and a direction perpendicular to the Pian Piccolo fault. A clear temporal trend of the aftershocks was then observed in the plot (Figure 7b). Therefore, the observed spatio-temporal pattern of aftershock migration can be interpreted by the seismologist to understand the earthquake mechanism and infer the possible controlling factor for the fault rupture. In this case, the diffusive-like aftershock temporal
trend can be further interpreted using a pore fluid source model. Through the interactive visualization operations, we can observe that aftershock propagation is consistent with a diffusive process and can also observe the spatial concentration of the aftershocks along the fault intersection. This information can be used to the derive the conclusion, as described in the work of *Walters et al.* [2018], that the intersecting structures act to channel the fluids and control the timing and order of the subsequent earthquakes.

**4. Conclusion**

We present a novel interactive approach and develop a prototype system to illustrate 3D aftershock point cloud that can help the geophysicist and seismologist to better understand the geometry of the complex fault system and the spatio-temporal pattern of the aftershocks. Following this approach, we design a set of interactive operations to facilitate the exploration of the data. Fast computation of LOF, 2D projection and plane fitting are implemented in addition to the visualization to reveal
additional, hidden information that is not obvious in the original data. Additionally, the point cloud can be visualized in many ways (e.g. 3D view, 2D plane projection, and animation), aiding viewers in interpreting the aftershock point cloud. A

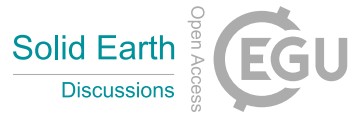

wide range of data filtering options (e.g. by magnitude, depth, location, time and LOF.) enable the users to avoid interference from invalid data, in order to focus on the useful information.

A case study of the 2016 Central Italy earthquake sequence shows that the proposed approach can complete a series of complex visualization operations in an user-friendly interactive manner. We can conveniently infer the fault geometry for both the main faults and the secondary faults, and clearly observe the spatio-temporal pattern of the fluid-driven aftershock migration. In future work, we plan to include more kinds of fault information (e.g. InSAR, GPS, source model, and fault scarp) into the visualization in order to to further facilitate the analysis of earthquake fault behavior.

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



**Figure 1: Flowchart of the proposed approach.**

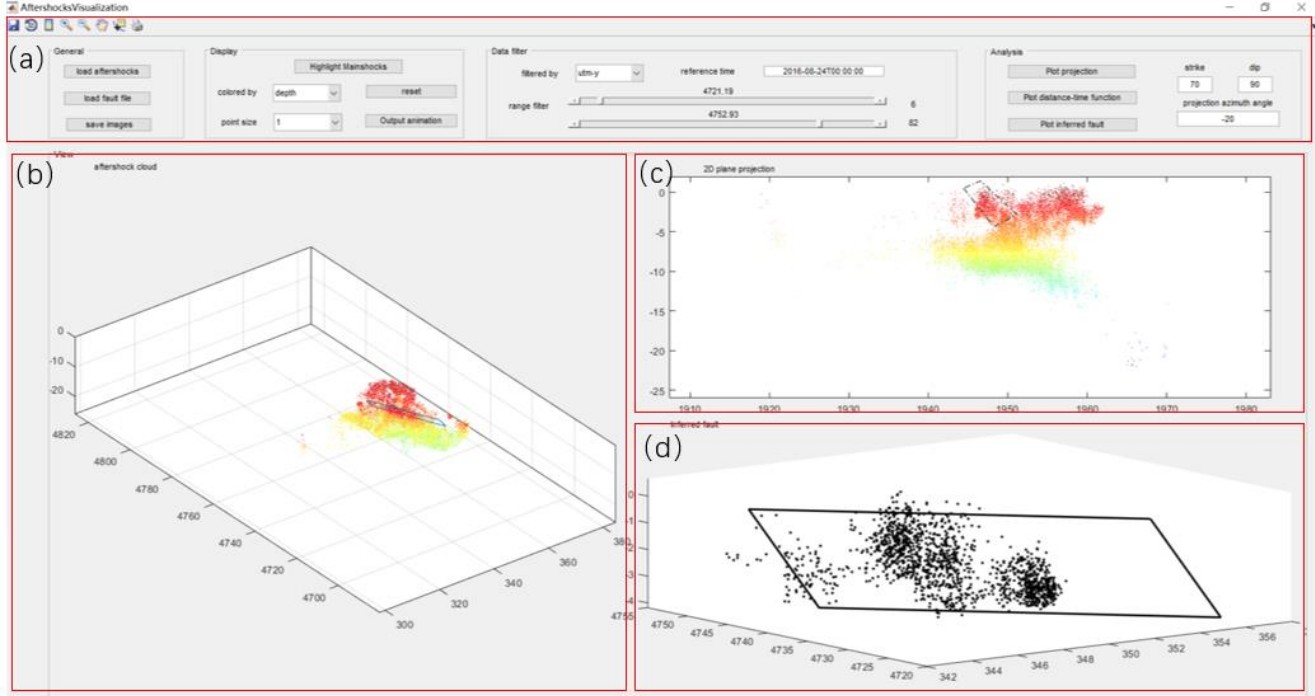

**Figure 2. Visual interface of our approach consisting of 4 interrelated components: (a) Operation tools, (b) 3D view, (c) 2D projection view, and (d) temporal view for plotting fitted fault plane or distance-time function.**



**Figure 3. An example of LOF scores for 2-D points**





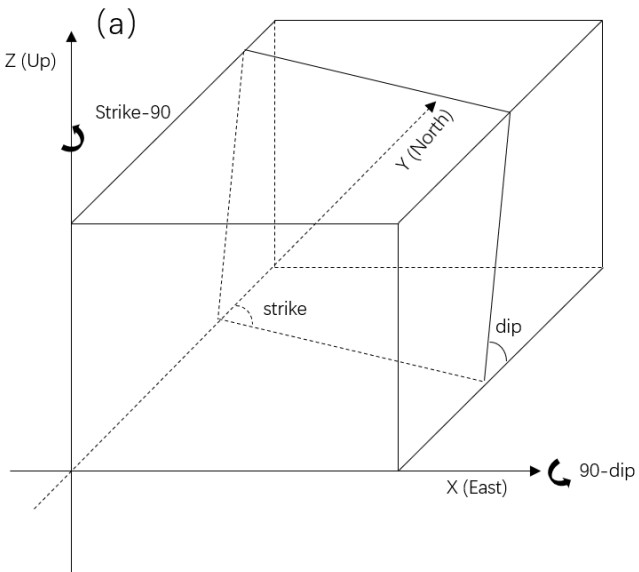
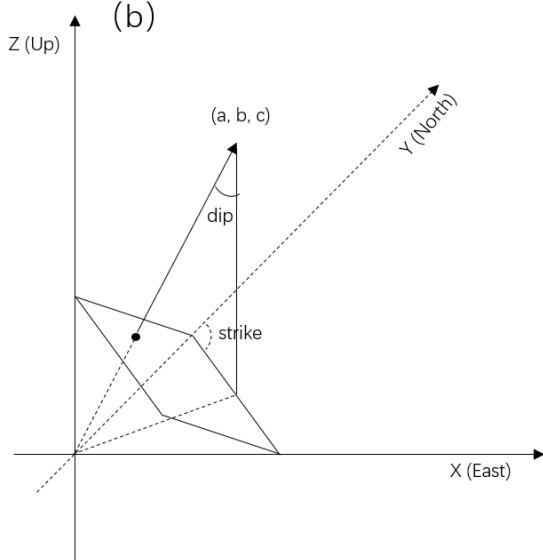

**Figure 4. (a) Illumination of projection to a plane defined by strike and dip. (b) Strike and dip parameters calculation from normal vector of a fitted plane.**





**Figure 5.** Seismotectonic setting of the Central Italy earthquake sequence. The three major earthquakes in the sequence are denoted by the red stars and beach ball symbols, and the two recent large historical events are colored in black. The dashed black rectangles represent the surface projection of the fault planes adopted in this study. The bold black lines are the two seismogenic normal fault systems, namely, the Mt. Vettore–Mt. Bove Fault (VBFS) system and the Mt. Gorzano Fault (GFS) system. The pink line shows the simplified trace of the preexisting compressional front named the Olevano-Antrodoco-Sibillini (OAS) thrust. Aftershocks are marked by the small dots with different colors. The blue, purple, and yellow points represent the events taking place between 2016.08.24–2016.10.24, 2016.10.24–2016.10.30, and 2016.10.30–2017.10.8, respectively.





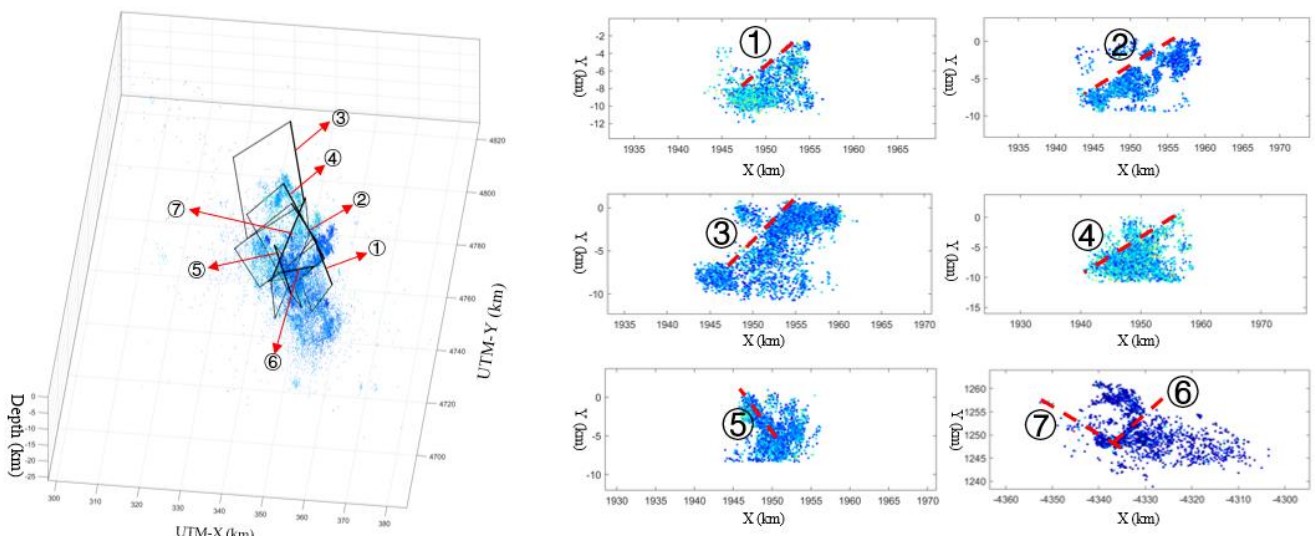

**Figure 6. Fault geometry identification and plane fitting.**





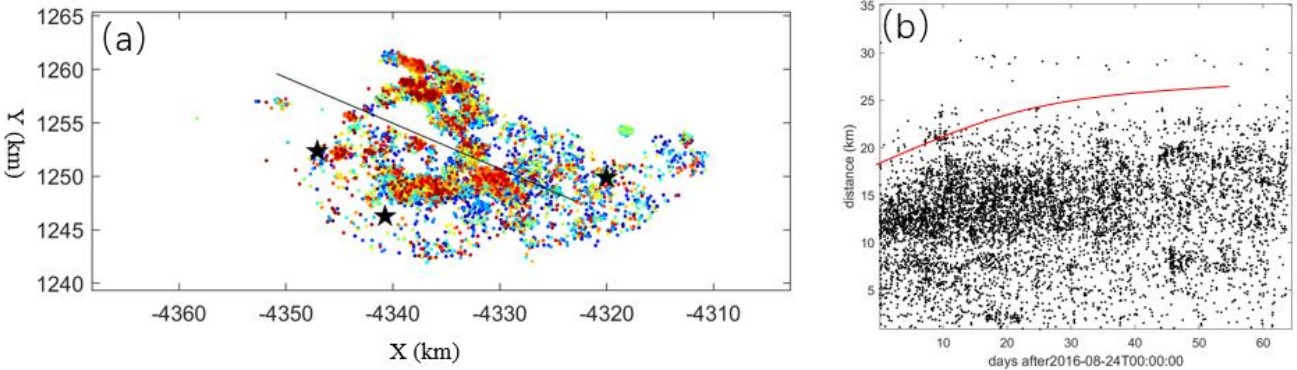

**Figure 7. Observation of aftershock migration. (a) Aftershock on the projection plane colored by the time. (b) Propagation distance as a function of time.**

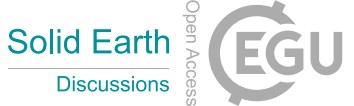

**Table 1. Geometry parameters of the inferred fault from the aftershock point cloud**

| Strike | Dip | UTM-X | UTM-Y | Length | Up-depth | Down-depth |
|--------|-------|--------|---------|--------|----------|------------|
| 159.06 | 55.64 | 357.36 | 4738.96 | 18.57 | 0.00 | 7.26 |
| 140.40 | 32.08 | 351.23 | 4752.59 | 15.76 | 0.00 | 9.31 |
| 166.93 | 35.92 | 346.80 | 4775.10 | 30.88 | 0.00 | 10.00 |
| 153.88 | 37.28 | 351.05 | 4753.25 | 29.86 | 0.00 | 7.09 |
| -19.12 | 60.71 | 348.21 | 4734.50 | 18.87 | 0.00 | 4.55 |
| -113.54 | 46.51 | 351.73 | 4734.97 | 8.45 | 0.00 | 11.17 |
| 7.81 | 85.33 | 348.16 | 4745.97 | 32.31 | 0.00 | 11.52 |