# Peer review of "Visual analytics of the aftershock point cloud data in complex fault systems"

_Solid Earth, 2019_

## Referee Comment (RC1) · Anonymous Referee #1 · 27 Jun 2019

The authors present a platform where the aftershock data can be visualized well, including the 3D view, fault plane fitting and outliers removal. Although the platform may be helpful to researchers that have crucial demands on 3D data visualization, it is not that clear why this platform is new and important in finding new fault planes or aftershock migrations. The proposed procedure of determining fault planes in principle relies heavily on the amount and accuracy of the catalog, which plays a much more important role. Below are some line comments: p1, line 24: ' where they are divided into a number of subparallel segments with the lengths of approximately 10-25 km ', is this true for all types of faults? p2, line 2, again, the complex fault geometry most directly results from an accurate catalog p2, lines 15-25, it is not clear why the interactive visual analysis represents an 'emerging field', I think it's only a good way to help

visually understand the data p2, line 16: assists -> assist p5, line 19, from point p (to) the k-th p8, line 7, any references for the triggering relationship? p9, line 12, is should -> should p10, line 3, the migration of aftershocks could also be related to afterslip and/or pressure transients p10, line 9, the northward trend is not obvious in Figure 7a

---

## Referee Comment (RC2) · Anonymous Referee #2 · 2 Jul 2019

Wang, et al. developed an interface to easily visualize aftershock distribution in a 3D setting. They produced a Matlab GUI that allows common processing and visualization techniques when interpreting aftershock temporal and spatial distributions.

In my opinion, this paper is peculiar as it does not produce any news scientific results, but offers a new computational tool to visualize and evaluate datasets. In that sense, I am not completely sure it fits within the traditional scope of SE. Now, we have to admit that a lot of the publications on data analysis only plot the data sets in different way to describe and understand them. So I understand the need for a simple tool to do so. I really appreciate that the tool if freely accessible and has 1) a GUI and 2) a GUI that is easy to use. The tool is still at a prototype stage but can be already used to start analyzing real dataset, as shown in the manuscript.

[Figure]

Although the main goal of such tool is to analyze complex aftershock sequence, I would advise the authors to not neglect the educational potential of such approach. Additionally, point clouds can also be used for analyzing tremor migration, and for tracking aseismic slip more generally. I would be interested to see how this tool could be used for tremors too.

Few comments: Page 1 line 25: "along the up-dip or along the down-dip" directions? Page 2 line 9 mainshock, not "the mains shocks" Page 9 line 12 should, not "is should" Figure 3 need labels on the x and y axes Figure 7 a colorbar would be useful for the time evolution The captions of Figures 3,6 and 7 should describe more what we see, so that we understand the approach and the plotted results while reading the caption.

———————————————————

---

## Author Comment (AC1) · 17 Jul 2019

Reviewer1: The authors present a platform where the aftershock data can be visualized well, including the 3D view, fault plane fitting and outliers removal. Although the platform may be helpful to researchers that have crucial demands on 3D data visualization, it is not that clear why this platform is new and important in finding new fault planes or aftershock migrations. The proposed procedure of determining fault planes in principle relies heavily on the amount and accuracy of the catalog, which plays a much more important role.

Authors: Thanks for this comment. It is true that the accuracy of aftershock data is of great importance in determining fault geometry and finding aftershock migrations.

[Figure]

However, the methods used for data analysis also play a very important role. It is widely accepted that visual analytic with interactive visual interfaces can amplify human cognitive capabilities, and is suitable for solving some complex problem relying on closely coupled human and machine analysis. When applying to the aftershock catalog, visual analytic can therefore help to discover some new fault planes or weak signals of aftershock migrations, which might be hard to observe directly from the data.

Specifically, the designed visual analytics procedure can assist the knowledge discovery from aftershock catalog in several ways: 1) by accelerating the fault segment discovery processing through 3D view manipulation (e.g. zooming, panning, and rotating); 2) by enhancing the recognition of fault segments pattern through rapid visual computing functions (e.g. interactive projection, plane fitting, and fault data fusing); 3) by reducing the influence from low-quality or irrelevant aftershock points through various filtering (e.g. depth, LOF, time, and magnitude filtering); 4) by enabling the aftershock migration exploration through providing more cognitive resources (e.g. animation, and propagation distance-time plotting).

Reviewer1: Below are some line comments: p1, line 24: ' where they are divided into a number of subparallel segments with the lengths of approximately 10-25 km ', is this true for all types of faults?

Authors: Thanks for the comment. The cited reference used normal fault dataset and is published 22 years ago. We will cite a more recent reference to update the description. Manighetti, I., C. Caulet, L. Barros, C. Perrin, F. Cappa, and Y. Gaudemer (2015), Generic along‐strike segmentation of Afar normal faults, East Africa: Implications on fault growth and stress heterogeneity on seismogenic fault planes, Geochemistry Geophysics Geosystems, 16(2), 443-467.

Reviewer1: p2, line 2, again, the complex fault geometry most directly results from an accurate catalog

Authors: Thanks for the comment. Given an accurate catalog, a proper analytic method

is also important for finding more details.

Reviewer1: p2, lines 15-25, it is not clear why the interactive visual analysis represents an 'emerging field', I think it's only a good way to help visually understand the data

Authors: Thanks for the comment. Visual analytics is considered as an outgrowth of the fields of data visualization [Pak Chung Wong and J. Thomas, 2004], which focuses on analytical reasoning facilitated by interactive visual interfaces. For better clarification, we down-phrase it from 'field' to 'research direction'.

Pak Chung Wong and J. Thomas (2004). "Visual Analytics". in: IEEE Computer Graphics and Applications, Volume 24, Issue 5, Sept.-Oct. 2004 Page(s): 20–21.

Reviewer1: p2, line 16: assists -> assist

Authors: Thanks for the comment. It would be corrected in the revised manuscript.

Reviewer1: p5, line 19, from point p (to) the k-th

Authors: Thanks for the comment. It would be corrected in the revised manuscript.

Reviewer1: p8, line 7, any references for the triggering relationship?

Authors: Thanks for the comment. We will include the following reference to support the triggering relationship. Papadopoulos, G. A., et al. "Earthquake triggering inferred from rupture histories, DInSAR ground deformation and stress-transfer modelling: the case of Central Italy during August 2016–January 2017." Pure and Applied Geophysics 174.10 (2017): 3689-3711.

Reviewer1: p9, line 12, is should -> should

Authors: Thanks for the comment. It would be corrected in the revised manuscript.

Reviewer1: p10, line 3, the migration of aftershocks could also be related to afterslip and/or pressure transients

Authors: Thanks for the suggestion. We will include the afterslip and/or pressure transients as the possible mechanisms for aftershock migration in the revised manuscript.

Reviewer1: p10, line 9, the northward trend is not obvious in Figure 7a

Authors: Thanks for the comment. The aftershock points in the north (left points on the figure) happened more lately (colored by red) than south. We will include north arrow and color bar to better demonstrate the northward trend of aftershock migration in the revised manuscript.

———————————————————

---

## Author Comment (AC2) · 17 Jul 2019

Reviewer2: Wang, et al. developed an interface to easily visualize aftershock distribution in a 3D setting. They produced a Matlab GUI that allows common processing and visualization techniques when interpreting aftershock temporal and spatial distributions.

In my opinion, this paper is peculiar as it does not produce any news scientific results, but offers a new computational tool to visualize and evaluate datasets. In that sense, I am not completely sure it fits within the traditional scope of SE. Now, we have to admit that a lot of the publications on data analysis only plot the data sets in different way to describe and understand them. So I understand the need for a simple tool to do so. I

really appreciate that the tool if freely accessible and has 1) a GUI and 2) a GUI that is easy to use. The tool is still at a prototype stage but can be already used to start analyzing real dataset, as shown in the manuscript.

Authors: Thanks for the comment. Visual analytics is considered as an outgrowth of the fields of data visualization [Pak Chung Wong and J. Thomas, 2004], which focuses on analytical reasoning facilitated by interactive visual interfaces. The visualization of seismic refraction data has already been widely applied to explore the distribution of petroleum and gas. However, to date, studies using the state-of-art visual analytics method to explore other kinds of seismic data (e.g. aftershock point cloud) to gain information about earthquakes have been lacking.

This study may be a little different from traditional earthquake research paper, because it focuses on visualization analytics method rather than traditional data processing algorithm. There are already a large number of automatic data analysis methods have been developed for earthquake data processing. However, the complex nature of earthquake problems makes it indispensable to include human intelligence at an early stage in the data analysis process. Visual analytics methods allow researchers to combine their human flexibility, creativity, and background knowledge with the data analysis capabilities of computer, to gain valuable knowledges from the complex earthquake data.

Our work proposes a novel interactive visualization method for exploring the 3D aftershock point cloud data. It is an attempt to test the potential of visual analytics methods in earthquake studies. The proposed visualization approach has been implemented in a software package (Aftershock Visualization [AFV1.0], https://github.com/caigenszu) and freely accessible to the scientific community. We believe this approach and the corresponding software package can benefit the scientific community for understanding complex fault system from aftershock data.

Reviewer2: Although the main goal of such tool is to analyze complex aftershock sequence, I would advise the authors to not neglect the educational potential of such

approach. Additionally, point clouds can also be used for analyzing tremor migration, and for tracking aseismic slip more generally. I would be interested to see how this tool could be used for tremors too.

Authors: Thanks for the comment. We will include the description for educational potential of this visualization analysis tool. We will also discuss the potential of this tool for analyzing tremor migration, and tracking aseismic slip in the revised manuscript.

Reviewer2: Few comments: Page 1 line 25: "along the up-dip or along the down-dip" directions?

Authors: Thanks for the comment. It would be corrected in the revised manuscript.

Reviewer2: Page 2 line 9 mainshock, not "the mains shocks" Page 9 line 12 should, not "is should"

Authors: Thanks for the comment. It would be corrected in the revised manuscript.

Reviewer2: Figure 3 need labels on the x and y axes Figure 7 a colorbar would be useful for the time evolution

Authors: Thanks for the comment. The colorbar will be included for Figure 7 in the revised manuscript.

Reviewer2: The captions of Figures 3,6 and 7 should describe more what we see, so that we understand the approach and the plotted results while reading the caption

Authors: Thanks for the comment. We will include more description in the caption in the revised manuscript.

---

## Editor Comment (EC1) · Caroline Beghein (Editor) · 18 Jul 2019

Dear authors,

Thank you for your responses to the reviewers comments. I will additionally suggest you include your detailed response to the first comment of reviewer 1 in your revised manuscript as other readers may have the same concerns. I would also like to see the reference to Wong and Thomas included in the manuscript for the same reason.

I also think your paper would be more suitable as a Method paper instead of a Research paper within Solid Earth since it does not present new research results in the traditional sense and is really focused on a visualization method. If you can change the type of paper you are submitting, please do so. Otherwise we will try to figure out how to

change it later. Thank you.

Sincerely,

Caroline Beghein

---

## Author Response (AR1)

We would like to thank the editor and reviewers for their constructive comments. We took the recommendations into account and revised our manuscript accordingly. In this file, the comments are marked in black, our responses are in red, and changes are in violet.
* * *
**Editor:**

(1) comment:

I will additionally suggest you include your detailed response to the first comment of reviewer 1 in your revised manuscript as other readers may have the same concerns. I would also like to see the reference to Wong and Thomas included in the manuscript for the same reason.

I also think your paper would be more suitable as a Method paper instead of a Research paper within Solid Earth since it does not present new research results in the traditional sense and is really focused on a visualization method. If you can change the type of paper you are submitting, please do so. Otherwise we will try to figure out how to change it later.

(2) response:

Thanks for this comment. Our response to the referee 1 was included in the revised manuscript. The reference to Wong and Thomas was added. In the new submission, we changed the type of paper to Method paper.

(3) changes:

P13, lines 21-30, P14, lines 1-5, detailed response to the first comment of reviewer 1 was included.

P3, line 7, a reference was cited to support it.

Our article type was changed to method paper.
* * *
**Referee 1:**

(1) comment:

The authors present a platform where the aftershock data can be visualized well, including the 3D view, fault plane fitting and outliers removal. Although the platform may be helpful to researchers that have crucial demands on 3D data visualization, it is not that clear why this platform is new and important in finding new fault planes or aftershock migrations. The proposed procedure of determining fault planes in principle relies heavily on the amount and accuracy of the catalog, which plays a much more important role.

(2) response:

Thanks for this comment. It is true that the accuracy of aftershock data is of great importance in determining fault geometry and finding aftershock migrations. However, the methods used for data analysis also play a very important role. It is widely accepted that visual analytic with interactive visual interfaces can amplify human cognitive capabilities, and is suitable for solving some complex problem relying on closely coupled human and machine analysis. When applying to the aftershock catalog, visual analytic can therefore help to discover some new fault planes or weak signals of aftershock migrations, which might be hard to observe directly from

the data.

Specifically, the designed visual analytics procedure can assist the knowledge discovery from aftershock catalog in several ways: 1) by accelerating the fault segment discovery processing through 3D view manipulation (e.g. zooming, panning, and rotating); 2) by enhancing the recognition of fault segments pattern through rapid visual computing functions (e.g. interactive projection, plane fitting, and fault data fusing); 3) by reducing the influence from low-quality or irrelevant aftershock points through various filtering (e.g. depth, LOF, time, and magnitude filtering); 4) by enabling the aftershock migration exploration through providing more cognitive resources (e.g. animation, and propagation distance-time plotting).

(3) changes:
P3, lines 7-11, some sentences were rephrased. And our article type was changed to method paper.
P13, lines 21-30, P14, lines 1-5, detailed response was included.
* * *
  (1) comment:
Below are some line comments:
p1, line 24: ' where they are divided into a number of subparallel segments with the lengths of approximately 10-25 km ', is this true for all types of faults?

(2) response:
Thanks for the comment. The cited reference used normal fault dataset and is published 22 years ago. We cited a more recent reference to update the description.
Manighetti, I., C. Caulet, L. Barros, C. Perrin, F. Cappa, and Y. Gaudemer (2015), Generic along-strike segmentation of Afar normal faults, East Africa: Implications on fault growth and stress heterogeneity on seismogenic fault planes, Geochemistry Geophysics Geosystems, 16(2), 443-467.

(3) changes:
P1, lines 27-30, a more recent reference was cited and the description was updated.
* * *
  (1) comment:
p2, line 2, again, the complex fault geometry most directly results from an accurate catalog

(2) response:
Thanks for the comment. Given an accurate catalog, a proper analytic method is also important for finding more details.

(3) changes:
P13, lines 21-30, P14, lines 1-5, detailed response was included.
* * *
  (1) comment:

p2, lines 15-25, it is not clear why the interactive visual analysis represents an 'emerging field', I think it's only a good way to help visually understand the data

(2) response:

Thanks for the comment. Visual analytics is considered as an outgrowth of the fields of data visualization [Pak Chung Wong and J. Thomas, 2004], which focuses on analytical reasoning facilitated by interactive visual interfaces. For better clarification, we down-phrase it from 'field' to 'research direction'.

Pak Chung Wong and J. Thomas (2004). "Visual Analytics". in: IEEE Computer Graphics and Applications, Volume 24, Issue 5, Sept.-Oct. 2004 Page(s): 20–21.

(3) changes:

P3, lines 7-8,, the sentence was rephrased and a reference was cited to support it.
* * *
  (1) comment:

p2, line 16: assists -> assist

(2) response:

Thanks for the comment. It was corrected in the revised manuscript.

(3) changes:

p2, line 28: corrected.
* * *
  (1) comment:

p5, line 19, from point p (to) the k-th

(2) response:

Thanks for the comment. It was corrected in the revised manuscript.

(3) changes:

P6, line 27: corrected.
* * *
  (1) comment:

p8, line 7, any references for the triggering relationship?

(2) response:

Thanks for the comment. We will include the following reference to support the triggering

relationship.

Papadopoulos, G. A., et al. "Earthquake triggering inferred from rupture histories, DInSAR ground deformation and stress-transfer modelling: the case of Central Italy during August 2016–January 2017." Pure and Applied Geophysics 174.10 (2017): 3689-3711.

(3) changes:
P10, line 1-2: a recent reference was cited to support the triggering relationship.
* * *
  (1) comment:
p9, line 12, is should -> should

(2) response:
Thanks for the comment. It would be corrected in the revised manuscript.

(3) changes:
P11, line 13: corrected.
* * *
  (1) comment:
p10, line 3, the migration of aftershocks could also be related to afterslip and/or pressure transients

(2) response:
Thanks for the suggestion. We included the afterslip and/or pressure transients as the possible mechanisms for aftershock migration in the revised manuscript.

(3) changes:
p12, lines 8-9: the afterslip and/or pressure transients were included as the possible mechanisms for aftershock migration
* * *
  (1) comment:
p10, line 9, the northward trend is not obvious in Figure 7a

(2) response:
Thanks for the comment. The aftershock points in the north (left points on the figure) happened more lately (colored by red) than south. We included north arrow and color bar to better demonstrate the northward trend of aftershock migration in the revised manuscript.

(3) changes:
P23: Figure 7 is replotted and the caption was rephrased to described the northward trend.
* * *
**Referee 2:**

(1) comment:

Wang, et al. developed an interface to easily visualize aftershock distribution in a 3D setting. They produced a Matlab GUI that allows common processing and visualization techniques when interpreting aftershock temporal and spatial distributions.

In my opinion, this paper is peculiar as it does not produce any news scientific results, but offers a new computational tool to visualize and evaluate datasets. In that sense, I am not completely sure it fits within the traditional scope of SE. Now, we have to admit that a lot of the publications on data analysis only plot the data sets in different way to describe and understand them. So I understand the need for a simple tool to do so. I really appreciate that the tool if freely accessible and has 1) a GUI and 2) a GUI that is easy to use. The tool is still at a prototype stage but can be already used to start analyzing real dataset, as shown in the manuscript.

(2) response:

Thanks for the comment. Visual analytics is considered as an outgrowth of the fields of data visualization [Pak Chung Wong and J. Thomas, 2004], which focuses on analytical reasoning facilitated by interactive visual interfaces. The visualization of seismic refraction data has already been widely applied to explore the distribution of petroleum and gas. However, to date, studies using the state-of-art visual analytics method to explore other kinds of seismic data (e.g. aftershock point cloud) to gain information about earthquakes have been lacking.

This study may be a little different from traditional earthquake research paper, because it focuses on visualization analytics method rather than traditional data processing algorithm. There are already a large number of automatic data analysis methods have been developed for earthquake data processing. However, the complex nature of earthquake problems makes it indispensable to include human intelligence at an early stage in the data analysis process. Visual analytics methods allow researchers to combine their human flexibility, creativity, and background knowledge with the data analysis capabilities of computer, to gain valuable knowledges from the complex earthquake data.

Our work proposes a novel interactive visualization method for exploring the 3D aftershock point cloud data. It is an attempt to test the potential of visual analytics methods in earthquake studies. The proposed visualization approach has been implemented in a software package (Aftershock Visualization [AFV1.0], https://github.com/caigenszu) and freely accessible to the scientific community. We believe this approach and the corresponding software package can benefit the scientific community for understanding complex fault system from aftershock data.

(3) changes:

P3, lines 7-11, some sentences were rephrased. And our article type was changed to method paper.
* * *
(1) comment:

Although the main goal of such tool is to analyze complex aftershock sequence, I would advise the authors to not neglect the educational potential of such approach. Additionally, point clouds can also be used for analyzing tremor migration, and for tracking aseismic slip more generally. I would be interested to see how this tool could be used for tremors too.

(2) response:
Thanks for the comment. We included the description for educational potential of this visualization analysis tool. We also discussed the potential of this tool for analyzing tremor migration, and tracking aseismic slip in the revised manuscript.

(3) changes:
P13, lines 12-13, P14, lines 9-11: The educational potential is mentioned and the potential of this tool for analyzing tremor migration, and tracking aseismic slip was included.
* * *
(1) comment:
Page 1 line 25: "along the up-dip or along the down-dip" directions?

(2) response:
Thanks for the comment. It would be corrected in the revised manuscript.

(3) changes:
Page 2 lines 1-2: corrected.
* * *
(1) comment:
Page 2 line 9 mainshock, not "the mains shocks" Page 9 line 12 should, not "is should"

(2) response:
Thanks for the comment. It was corrected in the revised manuscript.

(3) changes:
P11, line 13: corrected.
* * *
(1) comment:
Figure 3 need labels on the x and y axes Figure 7 a colorbar would be useful for the time evolution

(2) response:
Thanks for the comment. The labels were included for Figure 3 and the colorbar was included for Figure 7 in the revised manuscript.

(3) changes:
P20, P23: labels and colorbar were added.
* * *
(1) comment:

The captions of Figures 3,6 and 7 should describe more what we see, so that we understand the approach and the plotted results while reading the caption

(2) response:

Thanks for the comment. We will include more description in the caption in the revised manuscript.

(3) changes:

P20, P22, P23: more descriptions were included in the captions.

[revised manuscript text omitted]